# Long-Term Follow-Up in IgG4-Related Ophthalmic Disease: Serum IgG4 Levels and Their Clinical Relevance

**DOI:** 10.3390/jpm12121963

**Published:** 2022-11-28

**Authors:** Wei-Yi Chou, Ching-Yao Tsai, Chieh-Chih Tsai

**Affiliations:** 1Department of Ophthalmology, Zhongxing Branch, Taipei City Hospital, Taipei 103212, Taiwan; 2Institute of Public Health, National Yang Ming Chiao Tung University, Taipei 11221, Taiwan; 3Department of Business Administration, Fu Jen Catholic University, New Taipei City 24205, Taiwan; 4General Education Center, University of Taipei, Taipei 111036, Taiwan; 5Department of Ophthalmology, Taipei Veterans General Hospital, Taipei 11217, Taiwan; 6School of Medicine, National Yang Ming Chiao Tung University, Taipei 11221, Taiwan

**Keywords:** IgG4-related ophthalmic disease, IgG4-related disease, serum IgG4, corticosteroids

## Abstract

(1) Background: To analyze the association between long-term changes in serum IgG4 levels and the clinical course of patients with IgG4-related ophthalmic disease (IgG4-ROD). (2) Methods: Retrospective analysis of 25 patients with IgG4-ROD. (3) Results: Mean age at diagnosis was 60.68 years. Fifty-six percent of patients had bilateral ocular involvement and 32% had systemic associations. The ocular structures involved were the lacrimal gland (76%), orbital soft tissue (36%), extraocular muscle (20%) and infraorbital nerve (20%). According to last follow-up, 9 (36%) patients had normalized IgG4 levels, and 16 (64%) patients had elevated IgG4 levels. Patients with normalized IgG4 levels had better response to initial steroid treatment and attained a significantly lower IgG4 level after treatment (*p* = 0.002). The highest IgG4 levels were at baseline and disease recurrence, and lowest after initial treatment. At final follow-up, IgG4 levels differed in patients with remission (mean 326.25 mg/dL) and stable disease (mean 699.55 mg/dL). Subgroup analysis was performed in patients with remission, categorized according to whether IgG4 levels were normalized (9 patients) or elevated (10 patients) on last follow up. The elevated group had a higher percentage of bilateral disease, lacrimal gland involvement and recurrence. (4) Conclusions: IgG4-ROD patients with a greater response to initial steroid therapy were more inclined to have normalized IgG4 levels in the long term. Some patients remained in remission despite persistently elevated IgG4 levels, and had regular follow-up without treatment.

## 1. Introduction

IgG4-related disease is a fibroinflammatory clinicopathological entity, characterized by dense infiltration of lymphocytes and IgG4-positive plasma cells in various tissues and organs [1,2]. The disease has been described in numerous organ systems, including pancreas, bile duct, lacrimal glands, salivary glands, central nervous system, thyroid, gastrointestinal tract, kidney, retroperitoneum, lymph nodes, and skin [2]. The specific involvement of ocular and orbital tissues is collectively termed IgG4-related ophthalmic disease (IgG4-ROD) [3]. IgG4-ROD can manifest in lacrimal glands, orbital soft tissue, extraocular muscles, trigeminal nerve branches, and the supraorbital or infraorbital nerves [3,4]. IgG4-ROD may involve a single structure or multiple structures within the orbit, and may also exhibit bilateral disease [4]. The first-line treatment for active disease is systemic glucocorticoids. Immunosuppressive drugs and biologic agents, especially rituximab, have also been proven effective [4,5,6]. However, after initial steroid treatment, high recurrence rates (up to 61%) have been reported in the literature [7,8,9]. Therefore, certain patients may benefit from maintenance therapy [5]. Radiotherapy has also been used as additional therapy for recurrent or refractory IgG4-ROD [6,10].

IgG4-ROD is usually associated with an increase in serum IgG4 levels. The diagnostic criteria for IgG4-ROD includes imaging studies and histopathologic examinations of ophthalmic tissues, and blood tests showing elevated serum IgG4 (≥135 mg/dL), requiring at least two out of three criteria for diagnosis [3]. In addition to contributing to the diagnosis [11,12], serum IgG4 levels have been useful in determining disease activity and predicting relapse [13,14,15]. After steroid treatment, patients with IgG4-ROD responded well in the early phase with reduction in IgG4 levels [13,14,15]. Elevated serum IgG4 levels before [7] and after [15] steroid treatment have both been reported as risk factors for IgG4-ROD relapse. Despite these findings, the long-term serial changes of serum IgG4 levels are not well understood. The aim of this study was to determine the association between long-term changes in serum IgG4 levels and the clinical course of IgG4-ROD.

## 2. Materials and Methods

This research was designed as a single-institution, retrospective case series. Data collection and chart review were approved by the Institutional Review Board of Taipei Veterans General Hospital, Taipei, Taiwan (IRB-TPEVGH No.2021-10-002CC). The study was conducted in accordance with the tenets of the Declaration of Helsinki.

Records of patients with diagnosis of IgG4-ROD from April 2010 to April 2020 were examined. We applied the diagnostic criteria for IgG4-ROD by Goto et al. [3], as follows: (1) imaging studies showing enlargement of the lacrimal gland, trigeminal nerve, or extraocular muscle as well as masses, enlargement, or hypertrophic lesions in various ophthalmic tissues; (2) histopathologic examination showing marked lymphocyte and plasmacyte infiltration, and sometimes fibrosis; IgG4+ plasmacytes found and the ratio of IgG4+ cells to IgG+ cells of 40% or above, or more than 50 IgG4+ cells per high-power field; (3) blood test showing elevated serum IgG4 (≥135 mg/dL). Diagnosis was classified as “definitive” when (1), (2), and (3) were satisfied; “probable” when (1) and (2) were satisfied; and “possible” when (1) and (3) were satisfied. All study participants met inclusion criteria of the following: (a) diagnosis of definite, probable or possible IgG4-ROD, according to the aforementioned criteria [3]; (b) patients received treatment and had follow-up for at least 1 year. Subjects were excluded from the study if they had IgG4 lymphoma (n = 5), sclerosing orbital inflammation [16] (more aggressive clinical course, resulting in permanent visual loss, n = 1), follow-up less than 1 year (n = 5), or not fulfilling diagnosis of IgG4-ROD (only meeting one criterion, n = 4).

Anatomic locations of ocular lesions were evaluated both clinically and by reviewing orbital CT or MRI scans. Systemic involvement was determined by clinical manifestations and imaging studies by ultrasound, CT, MRI or PET/CT.

Collected data included demographics, laterality of ocular disease (unilateral or bilateral), systemic involvement of IgG4 disease, ocular structure of involvement (lacrimal gland, orbital soft tissue, extraocular muscle, infraorbital nerve, etc.), presence of recurrent disease, serological results (IgG level at baseline; IgG4 level at baseline, lowest level after initial steroid treatment, during recurrence, and last follow-up), duration of steroid treatment, and total follow-up time.

Recurrence was defined as symptom re-emergence or lesion enlargement after initial steroid treatment. Remission was defined as resolution of clinical symptoms and/or radiologic lesions. For the main outcome measures, patients were categorized into groups based on last follow-up condition: Remission with normalized IgG4 levels (group 1), remission with elevated IgG4 levels (group 2), and stable with disease. A cutoff value of 135 mg/dL was used. For additional analysis, the study population was alternatively grouped into those with unilateral or bilateral ocular involvement. 

Nonparametric tests were used for the following comparisons because most measurement data were not normally distributed. Categorical variables were analyzed using Fisher’s exact tests. Measurements were compared between groups 1 and 2, and unilateral and bilateral ocular involvement using Mann–Whitney *U* tests. Statistical significance was defined as *p* ≤ 0.05 and all analyses were conducted using SPSS statistical software (version 22, SPSS Inc., Chicago, IL, USA).

## 3. Results

A total of 25 patients diagnosed with IgG4-ROD were included in this study. On the basis of the diagnostic criteria for IgG4-ROD, the diagnosis was definite in 16 (64%) patients and possible in 9 (36%) patients. The demographics of the study population are summarized in Table 1. The mean age at diagnosis was 60.68 years. There were 17 (68%) men and 8 (32%) women. Ocular involvement was bilateral in 56% of patients. Systemic association was observed in 32% of patients. Disease involvement was isolated to one ocular structure in 60% of patients, while 28% and 12% of patients were concurrently affected in two and three ocular structures, respectively (not shown in table). The most common ocular structure involved was the lacrimal gland (76%), followed by orbital soft tissue (36%), extraocular muscle (20%) and infraorbital nerve (20%). Regarding disease course, 11 (44%) patients experienced recurrence during the follow-up period. The study population had initial steroid treatment of 14.53 ± 25.11 months (range 1.57 to 110 months), and follow-up period of 57.72 ± 33.36 months. According to last follow-up status, 9 (36%) patients had normalized IgG4 levels, and 16 (64%) patients had elevated IgG4 levels.

Figure 1 shows the serial changes in serum IgG4 levels from initial presentation to 60-month follow-up. The mean IgG4 level at baseline was the highest, with a steep decline to the lowest point after starting steroid treatment. After tapering and, for some patients, subsequently terminating steroid treatment, there was a gradual elevation in IgG4 levels. Later, the IgG4 levels fluctuated, and did not return to normalized levels in some patients. Table 2 shows comparison of patients with normal and elevated IgG4 levels according to last follow-up status; patients with normalized IgG4 levels had better response to initial steroid treatment and attained a significantly lower IgG4 level after treatment (*p* = 0.002). Table 3 further shows serum IgG4 levels at specific time points throughout the follow-up period, with the highest levels at baseline and disease recurrence (795.44 ± 553.10 mg/dL and 780.14 ± 568.38 mg/dL, respectively), lowest after initial treatment (176.36 ± 141.08 mg/dL), and 415.84 ± 345.87 mg/dL on last follow-up. At last follow-up visit, there was a distinct difference in IgG4 levels in patients with remission (mean 326.25 mg/dL) and those with stable disease (mean 699.55 mg/dL). In addition, in the 11 patients with recurrence, 3 had recurrent disease during tapering of steroids; as for the remaining 8 patients, the duration from discontinuing steroid treatment to recurrence was 11.92 ± 8.58 months (not shown in table).

After dividing the patients in remission into two groups depending on whether IgG4 levels were normalized (group 1) or elevated (group 2) on last follow up (Table 4), group 2 had a higher percentage of bilateral disease (70%), lacrimal gland involvement (90%) and recurrence (60%). In addition, throughout the disease course, group 2 had higher IgG4 levels at all time points documented: Baseline, lowest after initial treatment, and last follow up, albeit reaching statistical significance only at the lowest point (group 1, 82.19 ± 32.93 mg/dL vs. group 2, 152.80 ± 79.44 mg/dL, *p* = 0.043).

The study population was alternatively grouped into those with unilateral or bilateral ocular involvement (Table 5). All patients with bilateral ocular disease had involvement of the lacrimal gland, whereas only 45.5% of patients with unilateral disease were involved (*p* = 0.003). Patients with bilateral disease also had higher IgG and IgG4 levels at baseline, and a greater reduction in IgG4 levels after initial steroid treatment (*p* = 0.007, 0.004 and 0.005, respectively).

## 4. Discussion

We present a long-term follow-up of patients with IgG4-ROD focusing on the serial changes of serum IgG4 levels. To our knowledge, this case series of IgG4-ROD patients had the longest follow-up period compared with previous studies. In this cohort, up to 64% of patients had elevated IgG4 levels at last follow-up, which led us to investigate the correlation between IgG4 levels and their clinical significance.

Out of all patients in remission, the IgG4 levels failed to normalize in 10 of 19 patients (52.6%). The patients with elevated IgG4 concentrations had a relatively higher rate of bilateral disease, lacrimal gland involvement and recurrence. These patients also had higher IgG4 titers throughout the follow-up period, with statistically higher titers compared to the normalized group at the lowest point after initial treatment. These clinical and serological factors may indicate greater disease activity and poorer response to initial steroid treatment, rendering this group of patients prone to a higher IgG4 titer in the long term. Similarly, Wallace et al. [4] found the serum IgG4 level among those with bilateral disease to be higher compared to those with unilateral disease. Interestingly, serum IgG4 was higher in patients with lacrimal or salivary gland lesions compared to those with other affected organs [17,18], but no previous studies had compared serum IgG4 levels amongst different ocular structures involved, which is perhaps difficult because multiple ocular lesions can coexist. In a smaller case series of nine patients with IgG4-ROD and shorter follow-up time (average of 8.6 months), six patients had IgG4 levels at last follow-up, where two patients had clinical relapse, one patient had a normalized IgG4 level, and three patients had elevated IgG4 levels [14], but the case number may be too limited to come to comparable conclusions. A multicenter study on autoimmune pancreatitis showed that IgG4 levels failed to normalize in 63% of patients after treatment with oral steroids, but only 30% of these patients with persistent elevation of serum IgG4 levels had relapses [19]. We infer that IgG4-ROD patients with better response to initial steroid treatment would be more likely to resume normalized IgG4 levels in the long-term. In addition, despite persistent elevations of serum IgG4 levels, some patients can still remain in remission.

IgG4 is one of four subclasses of IgG, typically less than 5% of total IgG [20]. In general, IgG4 is a benign, non-pathogenic antibody. From a molecular structural standpoint, the disrupted C1q binding site of IgG4 contributes to its inability to activate the complement cascade [21]. The low affinity for classical Fcγ-receptors makes IgG4 inefficient in activating effector systems [21]. In addition, IgG4 forms small, non-precipitating immune complexes due to effective monovalency [22]. The pathophysiology of IgG4-related disease consists of multiple immune-mediated mechanisms, which remains to be fully elucidated. The counter-inflammatory properties of IgG4 and the fact that disease-specific IgG4 autoantibodies have not been found suggest that serum IgG4 is a response to inflammatory stimulus, more likely a consequence than a cause of the disease [23]. Two possible explanations for elevated IgG4 levels in patients in remission are “blocking antibody” and indicator of tolerance induction. IgG4 tends to appear after prolonged immunization with a decrease in symptoms. This is thought to be due to competition with IgE for allergen by IgG4, i.e., the blocking antibody, and inhibiting mast cell degranulation and/or preventing IgE-facilitated activation of T cells, and a suppression of the late-phase reaction [24,25]. Furthermore, IgG4 is a marker of tolerance induction. Usually at a single time point, the IgG4 levels in symptomatic patients are higher compared to those in asymptomatic patients. In long-term follow-up, a considerable increase in IgG4 can indicate that anti-inflammatory, tolerance-inducing mechanisms have been activated, in which inappropriate inflammatory reactions are reduced [20]. Therefore, regardless of IgG4 levels in patients with remission, we believe they can have regular follow up without intervention if they remain asymptomatic.

Approximately half of the study population had bilateral ocular lesions. These patients had statistically higher baseline IgG and baseline IgG4 levels compared to unilateral disease. This is in agreement with previous studies, which showed bilateral disease accounting for 49–62% of patients [26,27]. Wallace et al. [4] also found the baseline serum IgG4 level among those with bilateral disease to be statistically higher than unilateral disease. The reason for this finding is likely due to a higher disease activity in bilateral involvement. In addition, patients with bilateral disease also had better response to initial steroid therapy, resulting in a significantly greater reduction in IgG4 level. Interestingly, the change in serum IgG4 levels at recurrence (compared to baseline) differed between the two groups, with unilateral disease seeing a decrease in IgG4 values, and bilateral disease with an increase from baseline, albeit not reaching statistical significance. We can infer that there is a difference in the clinical course of unilateral and bilateral ocular lesions.

The lacrimal gland was the most common orbital structure affected in IgG4-ROD in our series and many previous studies [4,9,26,28,29]. Overall, the most frequently involved structure was the lacrimal gland (76%), followed by orbital soft tissue (36%), extraocular muscle (20%) and infraorbital nerve (20%). Although the percentage of ocular lesions varied with each study [4,9,26,28,29], our results were largely consistent with the results of a multicenter study by Goto et al. [28], which showed pathologic lesions in the lacrimal glands (86%), isolated and diffuse orbital lesions (19%), extraocular muscles (21%) and trigeminal nerve (20%). The high percentage of lacrimal gland involvement in IgG4-ROD may also explain why all patients with bilateral disease had lacrimal gland lesions in our study.

A few studies have looked into the treatment outcomes of IgG4-ROD. Regarding treatment regimens, an initial combination of glucocorticoid and immunosuppressant therapy had advantages over glucocorticoid monotherapy, with a longer relapse-free survival time [30], a lower relapse rate and a shorter glucocorticoid therapy duration [9]. Risk factors for relapse include multiple ocular lesions [9], extraocular muscle and/or trigeminal nerve enlargements [8], and additional radiotherapy after surgical debulking with oral steroids [27]. In the present series, with a mean follow-up duration of 57.72 months, 19 (76%) patients were in remission while 6 (24%) patients had stable disease. We infer that patients with IgG4-ROD have a favorable outcome, regardless of their serologic status in the long-term.

There are some limitations of the current study that should be addressed. First, the study is a retrospective, non-randomized controlled investigation, which might limit the generalizability of our findings. Second, nine patients (36%) did not receive biopsy and were diagnosed as “possible” IgG4-ROD. This is mainly due to difficulty in obtaining specimens in patients with extraocular muscle or deep orbital soft tissue involvement. Nevertheless, the findings of the present study have important implications for understanding the prognosis of IgG4-ROD.

## 5. Conclusions

Our results demonstrated that IgG4-ROD patients with greater response to initial steroid therapy, manifesting with lower serum IgG4 levels, would be more inclined to have normalized IgG4 levels in the long term. In addition, some patients can remain in remission despite persistently elevated serum IgG4 levels, and they can be followed up without treatment unless disease relapse. Moreover, different baseline serum IgG4 levels and disparities in disease course were observed between unilateral and bilateral ocular lesions. Future studies are needed to provide further insight into this complex disease.

## Figures and Tables

**Figure 1 jpm-12-01963-f001:**
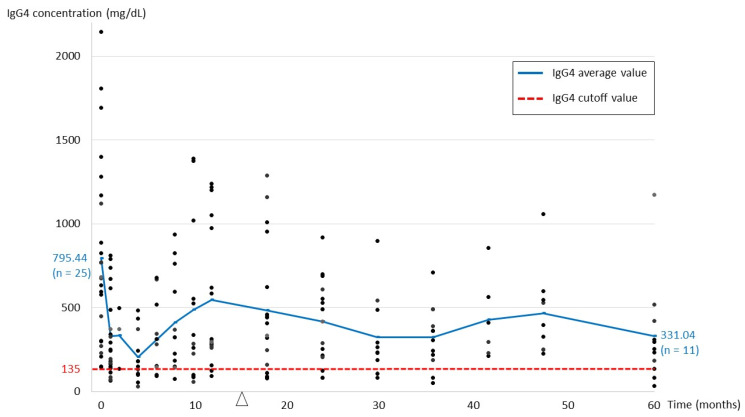
Serial changes in serum IgG4 levels from initial presentation to 60-month follow-up. The arrowhead indicates the end of initial steroid treatment (mean, 14.53 months).

**Table 1 jpm-12-01963-t001:** Baseline characteristics of patients with IgG4-related ophthalmic disease (n = 25).

Characteristic	Value
Age (mean, year)	60.68
Male gender, n (%)	17 (68%)
Bilateral, n (%)	14 (56%)
Systemic involvement, n (%)	8 (32%)
Ocular involvement	
Lacrimal gland, n (%)	19 (76%)
Orbital soft tissue, n (%)	9 (36%)
Extraocular muscle, n (%)	5 (20%)
Infraorbital nerve, n (%)	5 (20%)
Recurrence, n (%)	11 (44%)
Duration of steroid treatment (mean, month)	14.53
Follow-up time, month (mean, month)	57.72
Last follow-up status of IgG4 level *	
Normalized, n (%)	9 (36%)
Elevated, n (%)	16 (64%)

* For IgG4 levels at last follow-up visit, a cutoff value of 135 mg/dL was used to differentiate between normalized and elevated values.

**Table 2 jpm-12-01963-t002:** Comparison of patients with normal and elevated IgG4 levels according to last follow-up status.

Variable	IgG4 Normalized * (n = 9)	IgG4 Elevated * (n = 16)	*p* Value
Age (mean, year)	61.22	60.38	0.718
Male gender	77.8%	62.5%	0.661
Systemic involvement	44.4%	25%	0.394
Ocular involvement			
Lacrimal gland	55.6%	87.5%	0.142
Orbital soft tissue	44.4%	31.3%	0.671
Extraocular muscle	11.1%	25%	0.621
Infraorbital nerve	33.3%	12.5%	0.312
Recurrence	22.2%	56.3%	0.208
IgG level at baseline (mean, mg/dL)	1875.38	2227.13	0.636
IgG4 level at baseline (mean, mg/dL)	580.78	916.19	0.276
Lowest IgG4 level after initial steroid treatment (mean, mg/dL)	82.19	229.33	0.002
IgG4 level at last follow-up (mean, mg/dL)	78.50	605.60	0
Duration of steroid treatment (mean, month)	17.36	12.94	0.890

* For IgG4 levels at last follow-up visit, a cutoff value of 135 mg/dL was used to differentiate between normalized and elevated values.

**Table 3 jpm-12-01963-t003:** Serial changes in serum IgG4 levels at specific time points.

Variable	Value
IgG level at baseline (mean, mg/dL) (n = 24, 1 missing datum)	2104.78
IgG4 level at baseline (mean, mg/dL) (n = 25)	795.44
IgG4/IgG ratio at baseline (n = 24, 1 missing datum)	0.3586
Lowest IgG4 level after initial steroid treatment (mean, mg/dL) (n = 25)	176.36
IgG4 level during recurrence (mean, mg/dL) (n = 11)	780.14
IgG4 level at baseline in patients with recurrence (mean, mg/dL) (n = 11)	789.44
IgG4 level at last follow-up (mean, mg/dL) (n = 25)	415.84
In patients with remission (mean, mg/dL) (n = 19)	326.25
In patients with stable disease (mean, mg/dL) (n = 6)	699.55

**Table 4 jpm-12-01963-t004:** Comparison of patients in remission with normal and elevated IgG4 levels.

Variable	Group 1 (Remission with Normalized IgG4) ^1^ (n = 9)	Group 2 (Remission with Elevated IgG4) ^1^ (n = 10)	*p* Value
Last follow-up IgG4 (mean, mg/dL)	78.50	549.23	NA ^2^
Age (mean, year)	61.22	63.10	0.905
Male gender	77.80%	60%	0.628
Bilateral	33.30%	70%	0.179
Systemic involvement	44.40%	20%	0.35
Ocular involvement			
Lacrimal gland	55.56%	90%	0.141
Orbital soft tissue	44.44%	20%	0.35
Extraocular muscle	11.11%	30%	0.582
Infraorbital nerve	33.33%	20%	0.628
Recurrence	22.22%	60%	0.17
IgG level at baseline (mean, mg/dL)	1875.38	1857.67	0.888
IgG4 level at baseline (mean, mg/dL)	553.95	701.37	0.696
Lowest IgG4 level after initial steroid treatment (mean, mg/dL)	82.19	152.80	0.043
Reduction in IgG4 level (%) ^3^	75.58%	75.68%	0.847
Duration of steroid treatment (mean, month)	17.36	5.84	0.968

^1^ For IgG4 levels at last follow-up visit, a cutoff value of 135 mg/dL was used to differentiate between normalized and elevated values. ^2^ NA, not applicable. ^3^ Reduction in IgG4 level (%) = Absolute value of (Lowest level−Baseline level)/Baseline level × 100%.

**Table 5 jpm-12-01963-t005:** Comparison of parameters between patients with unilateral and bilateral ocular disease.

Variable	Unilateral (n = 11)	Bilateral (n = 14)	*p* Value
Age (mean, year)	63.00	58.86	0.291
Male gender	63.6%	71.4%	1
Systemic involvement	36.4%	28.6%	1
Ocular involvement			
Lacrimal gland	45.5%	100.0%	0.003
Orbital soft tissue	45.5%	28.6%	0.434
Extraocular muscle	18.2%	21.4%	1
Infraorbital nerve	9.1%	28.6%	0.341
Recurrence	44.4%	40.0%	1
IgG level at baseline (mean, mg/dL)	1649.55	2522.08	0.007
IgG4 level at baseline (mean, mg/dL)	455.09	1083.43	0.004
Lowest IgG4 level after initial steroid treatment (mean, mg/dL)	159.49	189.61	0.572
IgG4 during recurrence (mean, mg/dL)	205.80	1186.00	0.114
IgG4 level at last follow-up (mean, mg/dL)	281.98	521.02	0.183
Reduction in IgG4 level (%) ^1^	64.72	81.50	0.005
Change in IgG4 level at recurrence (%) ^2^	−8.21	21.93	0.667
Duration of steroid treatment (mean, month)	10.83	17.44	0.066

^1^ Reduction in IgG4 level (%) = Absolute value of (Lowest level−Baseline level)/Baseline level × 100%. ^2^ Change in IgG4 level at recurrence (%) = (Recurrence level−Baseline level)/Baseline level × 100%.

## Data Availability

Not applicable.

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
