# Peer review of "Long-Term Follow-Up in IgG4-Related Ophthalmic Disease: Serum IgG4 Levels and Their Clinical Relevance"

_jpm, 2022, doi:10.3390/jpm12121963_

Round 1

Reviewer 1 Report

In the study, to determine the association between long-term changes in serum IgG4 levels and the clinical course of IgG4-ROD, Chou and Tsai performed retrospective analyses of 25 patients. The results are presented in a necessary and sufficient manner, and the conclusions drawn from them are reasonable. IgG4-ROD is an intractable disease whose mechanism has not been fully elucidated, and these results focusing on IgG4 levels are valuable in estimating the prognosis of patients.  I consider the manuscript is appropriate to accept as present form.

Author Response

We thank the reviewer for their time, effort and kind comments.

Reviewer 2 Report

In this manuscript, the authors explored the association between long-term changes in serum IgG4 levels and the clinical course of patients with IgG4-related ophthalmic disease (IgG4-ROD). They compared the effect of steroid treatment between some groups: 1) patients with normal and elevated IgG4 levels according to the last follow-up status, 2) patients in remission with normal and elevated IgG4 levels, and 3) patients with unilateral and bilateral ocular lesions. The data showed IgG4-ROD patients with greater response to initial steroid therapy were more inclined to have normalized IgG4 levels in the long-term. Meanwhile, some patients remained in remission despite persistently elevated IgG4 levels and had regular follow-up without treatment. Overall, this study is interesting. I suggest acceptance for publication  After Minor Revisions:

1) In line 111 the minimum and maximum value of therapy period are welcome.

2) In Fig. 1 the legends of IgG4 concentration (mg/dL) vs time (months), on the “y” and “x” axes respectively, are missing. A label on the x-axis indicating the end of the mean of initial steroid treatment (14.53 months) is welcome. In the caption of Fig 1. put a label for the baseline (red) and the average value of IgG4 (blue line).

3) Table 2. In table 2 the last two values of P are missing.

4) Table 3. The first 2 values in Table 3 do not correspond to the values presented in Table 2. They must be  close to  2051.25 and 735.07, respectively.

5) Table 4.  I have inquietude on “Reduction in IgG4 level “. Please check the values, perhaps they are significantly lower. Consider calculate this value using the relative difference:

 100x((Baseline)  - (Lowest-Baseline ))/ Baseline

6) In lines, 211-219 authors comment:” Although serum IgG and IgG4 levels differed at baseline, no significant difference in long-term outcomes were observed between unilateral and bilateral ocular lesions.” However, the values of  IgG4 level at baseline are 455.09 and 1083.43(mean, mg/dL, Table 5)  for unilateral and bilateral, respectively. By comparing values of IgG4 during recurrence (205.80 and 1186.00 for unilateral and bilateral, respectively, Table 5); the unilateral ocular disease reduces 100x(455.09 - 205.80)/455.09=54.77%  while de bilateral reduces 100x(1186.00- 1083.43)/1186.00=  9.46%. This means that a significant effect on unilateral ocular disease is observed in long-term outcomes, while on the bilateral practically remains the same values. Please check this.

7) In lines 255-254 “Moreover, despite different baseline and recurrence serum IgG4 levels, no significant difference in long-term outcomes were observed between unilateral and bilateral ocular lesions.” It is partially correct according the estimations of Table 5. I see a significant reduction in unilateral ocular lesions.
